# Ovulation-Derived Fibronectin Promotes Peritoneal Seeding of High-Grade Serous Carcinoma Precursor Cells via Integrin β1 Signaling

**DOI:** 10.3390/cells15010080

**Published:** 2026-01-04

**Authors:** Che-Fang Hsu, Liang-Yuan Wang, Vaishnavi Seenan, Pao-Chu Chen, Tang-Yuan Chu

**Affiliations:** 1Center for Prevention and Therapy of Gynecological Cancers, Department of Medical Research, Hualien Tzu Chi Hospital, Buddhist Tzu Chi Medical Foundation, Hualien 970, Taiwan; cfhsu@tzuchi.com.tw (C.-F.H.); a0987090636@gmail.com (L.-Y.W.); vaishnaviseenan@gmail.com (V.S.); 2Department of Molecular Biology and Human Genetics, Tzu Chi University, Hualien 970, Taiwan; 3Institute of Medical Sciences, Tzu Chi University, Hualien 970, Taiwan; 4Department of Biotechnology, Vel Tech High Tech Dr. Rangarajan Dr. Sakunthala Engineering College, Chennai 600062, India; 5Department of Obstetrics & Gynecology, Hualien Tzu Chi Hospital, Buddhist Tzu Chi Medical Foundation, Hualien 970, Taiwan; coral.cpc@msa.hinet.net

**Keywords:** high-grade serous carcinoma (HGSC), follicular fluid, fibronectin, peritoneal metastasis, fallopian tube epithelium, integrin β1, ovulation

## Abstract

High-grade serous ovarian carcinoma (HGSC) is predominantly diagnosed at advanced stages with extensive peritoneal metastasis. A pivotal early event in HGSC development is the peritoneal seeding of tumor cells originating from the fallopian tube epithelial (FTE) precursor lesions. Ovulation releases follicular fluid (FF), which is known to contain oncogenic factors that promote FTE cell transformation. However, the specific mechanisms and factors within FF that drive the early metastatic seeding of precancerous FTE cells remain poorly defined. We investigated the role of FF in the peritoneal dissemination of FTE-derived cells, and the abundance of fibronectin (FN) as a potential key mediator. Functional assays were performed using FN-depleted FF to assess its impact on migration, invasion, anchorage-independent growth, and peritoneal attachment. The role of the fibronectin receptor, integrin β1 (ITGB1), and the signaling pathways were evaluated via knockdown studies. In vivo xenograft models were used to quantify peritoneal seeding, and mechanistic studies elucidated the involved signaling pathways. We identified FN as a critical component of FF, present at high concentrations (~210 µg/mL), that potently drives FTE cell migration, invasion, and peritoneal seeding. Depletion of FN from FF abrogated the majority of these pro-metastatic activities in vitro and led to a dramatic 82% reduction in peritoneal tumor seeding in vivo. Knockdown of ITGB1 similarly impaired seeding. Mechanistically, FF-derived FN activates the ITGB1/FAK-SRC signaling pathway to promote tumor cell motility and colonization. Our study establishes FF-fibronectin as an important regulator of the early peritoneal seeding of HGSC precursor cells. These findings reveal a direct link between ovulation and HGSC development, suggesting that targeting the FN-ITGB1 signaling axis may offer a novel preventive strategy for high-risk individuals.

## 1. Introduction

Ovarian high-grade serous carcinoma (HGSC) is a highly aggressive gynecological malignancy, with approximately 300,000 new cases and over 150,000 deaths reported annually worldwide. In 2020, HGSC accounted for 3.7% of global cancer cases and 4.7% of cancer-related deaths [1,2]. The poor survival rates associated with HGSC are largely due to late-stage diagnosis, as the disease is often asymptomatic in its early stages [3,4]. Recent advances in understanding the pathogenesis of HGSC have led to a consensus that the majority of cases originate from the secretory cells of the distal fallopian tube epithelium (FTE) [5,6,7,8,9]. A prevailing hypothesis suggests that precancerous cells in the FTE may already possess the ability to seed intraperitoneally [10,11,12], and these cells, once disseminated into the peritoneal cavity, can undergo further transformation, ultimately developing into HGSC [13,14,15].

Accumulating evidence has identified multiple oncogenic factors in ovulatory follicular fluid (FF)—such as reactive oxygen species (ROS), insulin-like growth factor 2 (IGF2), hepatocyte growth factor (HGF), and EGF-like ligands [16,17,18]—that drive the malignant transformation of FTE cells [16,17]. Beyond transformation, ovulation and the subsequent release of FF are implicated in the peritoneal dissemination of HGSC precursor cells. This is supported by in vivo studies where superovulation in a mouse model significantly enhanced the peritoneal seeding of transformed FTE cells compared to normal ovulation [19]. Further corroborating this, FF was shown to promote the attachment and growth of these cells on ex vivo peritoneal tissue in an AKT-dependent manner [14]. The translational relevance of these findings is underscored by studies of human peritoneal fluid (PF), where luteal-phase PF (which contains drained ovulatory FF) promotes anchorage-independent growth (AIG), invasion, and seeding of transformed FTE cells. In contrast, follicular-phase PF primarily supports earlier phenotypes like anoikis resistance and migration [15]. Together, these observations substantiate the “precursor escape” model, wherein precancerous FTE cells seed the peritoneum early, prior to full malignant transformation [11]. However, the specific FF-derived factors that orchestrate this initial escape and peritoneal seeding remain poorly defined.

Fibronectin (FN), the fourth most abundant protein in FF [20], engages integrins—notably integrin β1 (ITGB1)—to activate pro-tumorigenic pathways that drive cell adhesion, migration, and invasion [21,22,23]. Given the established role of FN-integrin signaling in cancer progression elucidating how ovulation-associated FN contributes to early tumor dissemination, could provide a translational foundation for preventing or treating ovarian cancer development [24,25].

In this study, we investigated the role of FN in FF-induced transformation of FTE cells. Our findings demonstrate that FF-derived FN plays a critical role in promoting migration, AIG, and peritoneal seeding of both partially and fully transformed FTE cells, highlighting its potential as a key mediator of early HGSC progression.

## 2. Materials and Methods

### 2.1. Cell Sources

In this study, cell lines represent various stages of FTE cell transformation. The FE25 cell line (P30), an immortalized human fimbrial epithelial cell line, was generated via transduction with human papillomavirus (HPV) E6/E7 and human telomerase reverse transcriptase (hTERT), representing a partially transformed FTE cell line [14]. The FEXT2 (p8) was derived from a xenograft tumor of FE25 cells transformed by human FF after intraperitoneally (i.p.) co-injection into NSG mice, representing a fully transformed FTE cell line [14,16]. All cell lines were maintained in MCDB105/M199 medium (1:1, Merck, Rahway, NJ, USA), supplemented with 10% fetal bovine serum (FBS, Thermo Fisher Scientific, Waltham, MA, USA) and penicillin/streptomycin (P/S, Corning Inc., Corning, NY, USA). The cell lines were provided by the Center for Prevention and Therapy of Gynecological Cancers, Department of Medical Research, Hualien Tzu Chi Hospital, Taiwan.

### 2.2. Clinical Specimens

FF was obtained during the oocyte retrieval process from women participating in the in vitro fertilization (IVF) program at Tzu Chi General Hospital in Taiwan, as previously documented [14,16]. Informed consent was secured from all participants involved in the study. The FF samples were subsequently pooled for the purpose of examining transformation activities. PF was collected from females of reproductive age during IVF surgery across two cohorts. As outlined in our prior research [16], PF was aspirated upon entry into the peritoneal cavity to minimize contamination from blood or tissue damage. Each sample underwent centrifugation to isolate the supernatant, which was then aliquoted and frozen prior to pooling and subsequent analysis. This investigation was received ethical approval from the Institutional Review Board of Tzu Chi Medical Center in Taiwan (Approval No. IRB110-238-A). The study adhered to all relevant guidelines and regulations, and a statement confirming that informed consent was obtained from all participants and/or their legal guardians is included in the manuscript. Research involving human participants was conducted in accordance with the principles of the Declaration of Helsinki.

### 2.3. ELISA Assay

FN levels in FF and PF were measured using a human FN ELISA kit (R&D Systems). Samples from IVF patients were centrifuged at 1000× *g* for 10 min, diluted 1:100, and applied to pre-coated 96-well plates. After 2 h of incubation at room temperature, the plates were washed, and biotinylated detection antibody was added for 1 h, followed by streptavidin-HRP incubation for 30 min. TMB substrate was applied, and the reaction was stopped with 2 N sulfuric acid. Absorbance was read at 450 nm, and FN concentrations were calculated based on a standard curve.

### 2.4. AIG

The AIG assay was modified for use in a 96-well plate format, as previously detailed [14]. In summary, sterilized agarose (Invitrogen, Carlsbad, CA, USA) was prepared in 50 mL tubes, melted, and maintained at 41 °C in a water bath prior to application. Each well was initially filled with a bottom layer of 0.8% soft agar in MCDB/M199 medium, followed by the addition of a top layer consisting of 0.4% soft agar mixed with 2000 cells. Every three days, the culture was supplemented with 20 µL of MCDB/M199 containing 10% fetal bovine serum (FBS). After a period of 14 days, colony counts were conducted in randomly selected fields at a magnification of 100×.

### 2.5. Anoikis Resistance Assay

The anoikis resistance assay was conducted utilizing a modified version of the CytoSelectTM 96-Well Anoikis Assay kit (Catalog Number CBA-081; Cell Biolabs Inc., San Diego, CA, USA) [14]. In summary, cells were seeded into 96-well plates that had been coated with agarose at a density of 2 × 10^3^ cells per well and were cultured in a serum-free medium. The culture medium was supplemented with 10% FF, 21 µg/mL FN, depleted FN FF or a control vehicle, with this supplementation repeated 48 h later. Following a 3-day incubation period, cell viability was assessed using the XTT colorimetric assay over a duration of 24 h. The results were normalized by subtracting the values obtained from cell-free controls.

### 2.6. 2D/3D Cell Migration and Invasion Assay

In the context of a two-dimensional migration assay, cells were cultivated in Micro-Insert 4 Well in µ-Dish (ibidi Catalog Number: 80466). Subsequently, 20,000 cells were plated in 6-well plates and allowed to incubate overnight. Each group was treated for 24 h and 100× microscopic images were taken. Microscopic images at a magnification of 100× were obtained for each sample and drug group after a 24 h period. The software Image J (National Institutes of Health, Bethesda, MD, USA) was employed to quantify the image area for analysis. Comparisons among the groups were conducted following the subtraction of the migration area measured at the 0 h mark. For 3D Cell migration was assessed utilizing a 24-well transwell insert (Corning Inc., Corning, NY, USA). Cells were initially seeded in the upper chamber at a density of 2 × 10^4^ cells in 0.3 mL of serum-free MCDB/M199 media. Following a 24 h incubation at 37 °C in a 5% CO_2_ atmosphere, 0.5 mL of medium supplemented with 10% fetal bovine serum (FBS) was introduced to both the upper and lower wells. Subsequently, the membranes were fixed in 4% paraformaldehyde for 20 min, and the cells that migrated to the lower surface were stained using Giemsa. For the invasion assay, the transwell insert were coated with 60 µL of diluted Matrigel (Corning Inc.) and allowed to incubate overnight. The upper chamber was then populated with 1 × 10^4^ cells. After a 48 h period, the number of cells that migrated to the lower chamber was quantified by counting in three random fields per filter.

### 2.7. Western Blot Analysis

Briefly, a total of 40 μg of protein from the sample was subjected to separation via 10% sodium dodecyl sulfate polyacrylamide gel electrophoresis (SDS-PAGE). Following the electrophoretic separation, the gels were transferred onto Immobilon-P membranes (EMD Millipore, Burlington, MA, USA). The polyvinylidene fluoride (PVDF) membranes were subsequently blocked with a solution of 5% (*w*/*v*) non-fat milk powder in phosphate-buffered saline (PBS) containing 0.1% Tween-20 (PBST) at room temperature for one hour. The membranes were then incubated with the primary antibody at 4 °C overnight. After washing with PBST, the blots were treated with a secondary antibody in PBST at room temperature for one hour. Detection of the antibodies was achieved using a chemiluminescent horse radish peroxidase (HRP) detection reagent (EMD Millipore, Burlington, MA, USA). Quantitative analysis of the target protein was conducted utilizing Image J software, 1.54p. A comprehensive list of all antibodies employed in this study is provided in Appendix A.

### 2.8. Xenograft Tumor Model

NOD/Shi-scid/IL-2Rγnull (NSG) mice were utilized for the xenograft tumor-seeding assay. In this study, female mice aged 7 to 8 weeks were administered an intraperitoneal injection of 1 × 10^5^ FE25-FLUC cells suspended in 200 µL of either PBS or 10% FF. The FEXT2-LUC cell line was established from a xenograft tumor of FE25 cells that had been modified through transduction with the luciferase-expressing lentivirus pLAS3w.FLuc.Puro, sourced from the National RNAi Core Facility of Academia Sinica in Taiwan (https://rnai.genmed.sinica.edu.tw/ (accessed on 10 September 2025). The fluid injection was repeated three days post-initial injection, and on the fifth day, signals within the peritoneal cavity were detected and quantified using an in vivo imaging system. Bioluminescence signals were analyzed using Living Image® software (PerkinElmer, Shelton, CT, USA). During IVIS detection, 1–2% Isoflurane was used to maintain anesthesia for imaging. Mice were euthanized using CO_2_ asphyxiation in accordance with ethical guidelines. All experimental procedures involving mice were sanctioned by the Animal Care and Use Committee of Tzu Chi University (Approval ID: 110-35), with the animal models provided by the Animal Center of Hualien Tzu Chi University.

### 2.9. Statistics

All data are presented as mean ± standard error. Statistical analyses were performed using GraphPad Prism version 8.0 (GraphPad Software, La Jolla, CA, USA) and Microsoft Office Excel 2019 (Microsoft, Redmond, WA, USA). Detailed information on statistical analysis is described in figure legends. All data generated or analysed during this study are included in this published article [and its Appendix A].

## 3. Result

### 3.1. High Molecular Weight Components, Including FN, in FF Promote Migration of HGSC Precursor Cells

To investigate the role of FF in promoting the metastatic behavior of FTE cells, we utilized two established FTE cell lines: FE25 (partially transformed, representing early-stage precursor cells) and FEXT2 (fully transformed, representing late-stage precursor cells adapted to the peritoneal environment) [16,19]. Both cell lines exhibited enhanced metastatic phenotypes, including migration, anoikis resistance, peritoneal attachment, and invasion, in response to FF treatment [14]. To identify the specific factors in FF responsible for these effects, we fractionated FF based on molecular weight. We found that the >100 kDa fraction, but not the <100 kDa fraction, significantly promoted the migration of FE25 cells (Figure 1A,B).

Among the high molecular weight components in FF, we identified FN, the fourth most abundant protein in FF [20], as a key candidate. Analysis of paired FF and PF samples from 25 women undergoing oocyte retrieval revealed that FF contained 2.95-fold higher FN levels than PF (210 ± 76 μg/mL vs. 71 ± 25 μg/mL). This concentration was approximately 55% of that found in malignant ascites samples (385 ± 55 μg/mL, *n* = 8) (Figure 1C). At a concentration equivalent to 10% FF (21 μg/mL), recombinant FN promoted the migration of FE25 and FEXT2 cells to 69% and 40% of the levels induced by FF, respectively (Figure 1A, Table 1). These findings suggest that FF-derived FN contributes to the migration of transforming FTE cells in the peritoneal cavity, even in non-malignant states.

### 3.2. Depletion of FN from FF Partially Compromises Migration and Invasion

To further elucidate the role of FN in FF-mediated transformation, we depleted FN from FF using immunoprecipitation. In 2D migration assays, FF and recombinant FN increased the migration of FE25 cells by 7.95-fold and 5.50-fold, respectively, and FEXT2 cells by 2.40-fold and 1.83-fold, respectively. FN depletion reduced the migration-promoting effects of FF by 54% in FE25 cells and 20% in FEXT2 cells compared to IgG controls (Figure 2A). Similarly, in 3D transwell migration assays, FF and recombinant FN increased migration by 6.47-fold and 4.12-fold in FE25 cells and 7.23-fold and 6.13-fold in FEXT2 cells, respectively. FN depletion resulted in a 73% reduction in FE25 cell migration and a 44% reduction in FEXT2 cell migration (Figure 2B, Table 1).

In Matrigel invasion assays, FF significantly enhanced the invasive capacity of both cell types. FN depletion reduced this effect by 66% in FE25 cells and 36% in FEXT2 cells, indicating that FN plays a more prominent role in the invasiveness of partially transformed than the fully transformed FTE cells. Interestingly, recombinant FN alone had minimal effects on FE25 cell invasion, suggesting that additional factors in FF, such as HGF/cMET signaling [26], may synergize with FN to drive invasion. In contrast, FEXT2 cells exhibited a robust invasive response to recombinant FN (Figure 2C, Table 1), highlighting differences in the dependency on FN between precancerous and cancerous HGSC cells.

### 3.3. FF Enhances Proliferation Independently of FN

We next assessed the proliferative effects of FF and FN on FE25 and FEXT2 cells. FF increased cell proliferation by 58% in FE25 cells and 43% in FEXT2 cells. However, recombinant FN had no significant effect on proliferation, and FN depletion did not alter the proliferative effects of FF (Appendix A).

### 3.4. FN Plays a Minor Role in Anoikis Resistance and Anchorage-Independent Growth (AIG)

FF significantly enhanced anoikis resistance in both cell lines, increasing cell survival by 38%. Recombinant FN, however, only marginally increased anoikis resistance in FEXT2 cells (10%) and had no effect in FE25 cells (Figure 3A). Similarly, in AIG assays, FF treatment resulted in the formation of 7 colonies in FE25 cells and 15 colonies in FEXT2 cells, whereas recombinant FN did not promote colony growth. FN depletion reduced colony formation by 54% in FE25 cells and 68% in FEXT2 cells (Figure 3B). These results suggest that while FF strongly enhances anoikis resistance and AIG, FN contributes minimally to these effects, implicating other FF components as the primary drivers.

### 3.5. ITGB1 Knockdown Reduces FF-Induced Migration but Not Invasion

To investigate the role of ITGB1, the primary FN receptor, in FF-mediated transformation, we performed RNA interference to knockdown ITGB1 expression in both cell lines (Appendix A). ITGB1 knockdown reduced FF-induced 2D migration by 71% in FE25 cells and 40% in FEXT2 cells (Figure 4A). Similarly, in 3D migration assays, ITGB1 knockdown decreased FF-induced migration by 59% in FE25 cells and 20% in FEXT2 cells (Figure 4B). However, ITGB1 knockdown did not significantly affect FF-promoted invasion in Matrigel assays (Figure 4C). The results indicate that ITGB1 partially mediates the pro-migratory effects of FF but not its pro-invasive activity.

### 3.6. ITGB1 Knockdown Compromises Peritoneal Attachment and AIG

ITGB1 knockdown also impaired FF-induced peritoneal attachment and AIG. Specifically, ITGB1 knockdown reduced the attachment growth of FE25 cells to ex vivo cultured peritoneum by 81% compared to sh-Luc controls (Figure 5A). Similarly, FF-induced AIG colony formation was abolished in both FTE cells with ITGB1-knockdown (Figure 5B). Table 2 summarizes these changes in FF treatment with or without ITGB1 knockdown.

### 3.7. Pro-Metastasis Effect of the FF-FN/ITGB1 Signaling

To investigate the pro-metastatic activity of FF-FN and the associated ITGB1 signaling, we performed intraperitoneal (i.p.) xenografts of luciferase-expressing FE25 cells in NSG mice (Figure 6A). Mice received twice-weekly injections of either FN-depleted FF (de-FN) or control IgG-depleted FF (de-IgG). While the de-FN group showed a 61% reduction in bioluminescent signal by day 5, this difference was not statistically significant (*p* = 0.1) due to high variation within the control group (Figure 6B,C). However, by day 46, a significant and pronounced 82% reduction in tumor burden was observed in the de-FN group (*p* < 0.005), demonstrating that FN is critical for efficient peritoneal seeding and outgrowth.

In a complementary experiment, we xenografted FE25 cells with ITGB1 knockdown to directly target the FN receptor (Figure 6D). To maximize signal detection sensitivity, we quantified the maximum number of photon events. The ITGB1-knockdown group exhibited a significant 30% reduction in seeding on day 12 (*p* < 0.005) (Figure 6E,F). This suppression became more substantial by day 40, with a 55% reduction in signal intensity; however, high variability in the control group precluded statistical significance at this later time point. The observed heterogeneity in tumor growth across all experiments may be attributable to the inherent chromosomal instability of FE25 cells, which harbor p53 and Rb disruptions and acquire progressive copy number variations [16].

### 3.8. FF-FN Activities Were Partially Mediated by ITGB1-FAK/SRC/AKT Pathways

Compared to FF treatment, which significantly increased the phosphorylated (p) FAK, SRC and AKT, FN treatment in both FTE cell lines predominantly enhanced p-FAK and p-SRC, with a mild increase in p-AKT. The effects of ITGB1 knockdown on these signaling proteins were more complex. We observed a significant increase in p-AKT and p-SRC levels in FE25 and FEXT2 cells cultured in serum-free conditions, respectively. As expected, ITGB1-knockdown in FE25 cells resulted in a diminished response to FN treatment, specifically in the phosphorylation of FAK and SRC. A similar, albeit less pronounced, trend was observed in ITGB1-knockdown FEXT2 cells (Figure 7). These findings suggest the involvement of FN/ITGB1/FAK-SRC and FN/ITGB1/AKT signaling pathways in both FE25 and FEXT2 cells (Figure 8B).

## 4. Discussion

This study elucidates the critical role of FF-FN in promoting the early peritoneal seeding of HGSC precursor cells from the fallopian tube epithelium. We demonstrate that FF-derived FN enhances the migration, invasion, and anchorage-independent growth of both partially transformed (FE25) and fully transformed (FEXT2) FTE cells, primarily through the FN/ITGB1/FAK-SRC signaling axis. These findings provide mechanistic insights into how ovulation-associated factors contribute to the early metastasis of HGSC and highlight potential therapeutic targets for prevention.

### 4.1. FF-Derived FN as a Key Mediator of Early Metastasis

Our results establish FN, a major constituent of FF, as a critical promoter FTE cell migration and invasion. Fractionation studies pinpointed the >100 kDa fraction of FF, which contains FN (220–250 kDa), as the primary driver of these pro-metastatic behaviors. The potency of FN is underscored by its abundance, with concentrations in FF reaching 210 μg/mL—three-fold higher than levels found in PF. Functional depletion of FN demonstrated that it is responsible for the majority of FF-induced migration, invasion, and AIG in FTE cells. It is important to note that other transformative phenotypes such as cell proliferation, anoikis resistance, and peritoneal attachment remained largely unaffected by FN depletion (Appendix A). The in vivo relevance of these findings was confirmed by xenograft models, where FN depletion resulted in a striking 82% reduction in the peritoneal seeding of precursor cells, underscoring FN’s pivotal role in the early dissemination of HGSC.

While FN is a principal effector, it acts within a collaborative network of other FF-derived factors that contribute to the metastatic potential of FTE cells. For instance, hepatocyte growth factor (HGF) and the proteoglycan versican have been shown to enhance migration, invasion, and seeding [27]. The ovarian hormone activin A also significantly stimulates FTE cell migration [28], and vitronectin can support the adhesion and proliferation of these cells on non-adherent surfaces [28]. A comprehensive proteomic analysis of human FF identified 742 proteins, with FN ranking as the fourth most abundant, followed by vitronectin (#31), and versican (#88) [20]. Notably, activin A, which is present at functionally active concentrations (1.7–267.9 ng/mL) [29], was not detected in this proteomic analysis. In conclusion, our data position FN as the most abundant and predominant factor in FF responsible for driving the migration, invasion, and peritoneal seeding of FTE cells, key early events in HGSC pathogenesis.

Meanwhile, VCAM-1—another ligand for integrin α4β1 (a β1 heterodimer)—may further contribute to the prometastatic activity, as it was also identified in FF (ranking 170 out of 366 proteins) [20]. As a primary adhesion molecule, VCAM-1 is well-known for mediating firm, static cell adhesion, such as in leukocyte binding to the endothelium [30]. It is likely VCAM-1 present in FF contributes to the enhanced peritoneal attachment of HGSC precursor cells observed in our in vitro experiments and promotes their peritoneal seeding in vivo.

### 4.2. Role of ITGB1 Signaling in Promoting FTE Cell Migration and Tumor Progression

This study investigated the role of ITGB1 in FF-induced transformation of FTE cells. We found that ITGB1 knockdown significantly reduced the migration-promoting effect of FF, although it did not affect the invasion capabilities of the cells. Additionally, ITGB1 knockdown abolished the ability of the cells to attach and grow on the peritoneum, even in the presence FF. Furthermore, the knockdown moderately decreased the AIG-promoting effect of FF. Integrin α5β1 is the most well-characterized receptor for FN and plays a crucial role in mediating cell migration and adhesion [31,32]. In addition to FN, integrin α5β1 also binds to fibrinogen and fibrillin, which ranked #88 and #406, respectively, in the list of 742 identified proteins in FF [20]. When combined with other alpha integrins, integrin β1 serves as a receptor for numerous extracellular proteins. This broad binding capacity may explain why the knockdown of ITGB1 resulted in more extensive phenotypic changes and a greater compromise of the effects induced by FF exposure.

### 4.3. Fully Transformed FTE Cells Are Less Dependent on FF Signals for Migration and Invasion

This study found that the fully transformed FEXT2 cells exhibit reduced dependence on FF-FN for both 2D and 3D cell migration compared to the strong dependence observed in the non-transformed FE25 cells. It is likely that, similar to other ovarian cancer cells like SKOV3 [33], fully transformed and peritoneal metastatic FTE cells have established autocrine signaling pathways involving receptor tyrosine kinase signals such as HGF/cMET [34]. This autocrine signaling cooperates with FN/integrin interactions to enhance cell invasion [35,36], allowing these cells to become less reliant on the exogenous FN and HGF signals provided by FF.

### 4.4. Signals Downstream of FF-FN/ITGB1

Signal pathways downstream of FN/ITGB1 include FAK-SRC/ERK, which mediates cell migration and adhesion, and FAK-SRC/AKT, which mediates cell survival [37]. Studies of phosphorylated signaling proteins in FTE cells treated with FF, FN-depleted FF, and pure FN support these dual functions. They demonstrate that the FF-FN-ITGB1 signaling pathway is primarily mediated through FAK-SRC and FAK-SRC/AKT, exerting effects on migration and peritoneal attachment as well as promoting AIG, respectively (Figure 8). Interestingly, while depletion of FN from FF significantly impacted the invasion of FF-treated FEXT2 cells, pure FN alone had no effect on cell invasion. This suggests that FN may positively interact with other pro-invasion signals promoted by FF [38]. Previously, we have found FF harbors coagulation cascade proteins which activated HGF and exert pro-invasion and pro-AIG activities to the exposed FTE cells [17]. Several studies have also demonstrated the formation of complexes between c-Met and integrins such as α5β1, α3, and α6β4 in tumor cells, both in the presence and absence of HGF [34,39,40]. Functionally, these interactions drive cell migration, invasion, anchorage-independent survival, and ultimately metastasis in cancer [41,42]. Moreover, HGF can bind to integrin ligands like FN, enhancing c-Met-integrin cooperation and promoting cell migration, proliferation, and survival, particularly during cancer progression [40].

## 5. Conclusions

In conclusion, this study demonstrates that FN in FF promotes the migration, adhesion, and anchorage-independent growth of HGSC cells. During ovulation, these FN-driven activities act upon exposed precancerous and cancerous cells in the fallopian tube, facilitating their peritoneal seeding and metastasis (Figure 8A). This newly identified pro-metastatic function of FF further underscores the overwhelming oncogenic role of ovulation in ovarian cancer pathogenesis. Consequently, targeting the FN-ITGB1 axis may present a promising therapeutic strategy to prevent the early metastasis of fallopian tube-derived precursor cells.

## Figures and Tables

**Figure 1 cells-15-00080-f001:**
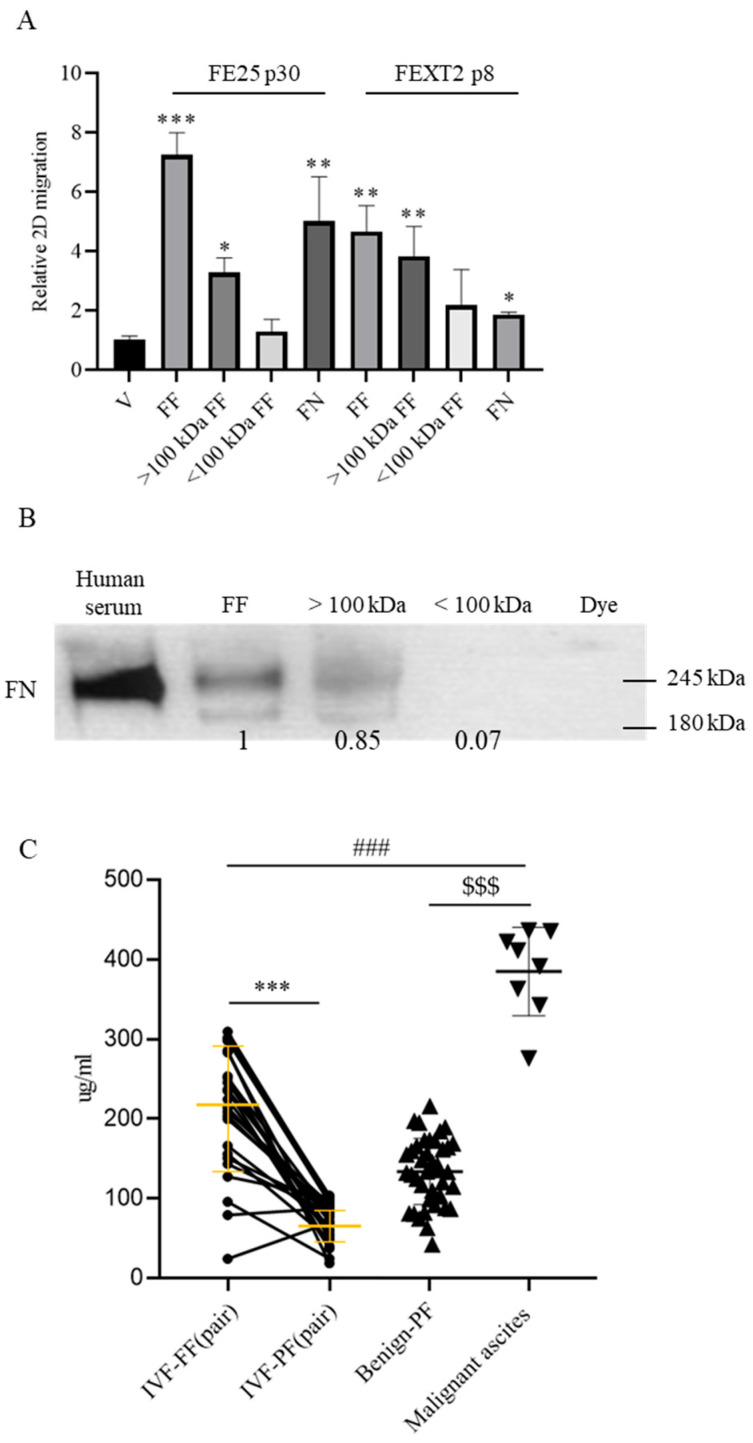
Migration of FTE cells is promoted by high molecular weight fraction of FF and by FN which is abundantly present in FF. (**A**) The 2D cell migration assay was performed on partially (FE25) and fully (FEXT2) transformed FTE cells for 48 h under serum-free conditions. FF: 10% follicular fluid; >100 kDa and <100 kDa: centrifuge filter isolates of 10% FF; FN: recombinant fibronectin at a concentration equivalent to 10% FF (21 μg/mL); * *p* < 0.05, ** *p* < 0.01, *** *p* < 0.001, as compared with the vehicle control. data were mean ± SD from triplicate experiments (by two-tailed, unpaired Student’s *t*-test). (**B**) The density of FN were measured using Western blotting. Dye: Vehicle only. (**C**) ELISA analysis of clinical samples for FF/PF. Yellow bars indicate the mean ± SD; triangles represent benign or malignant ascites; ***/###/$$$ *p* < 0.001.

**Figure 2 cells-15-00080-f002:**
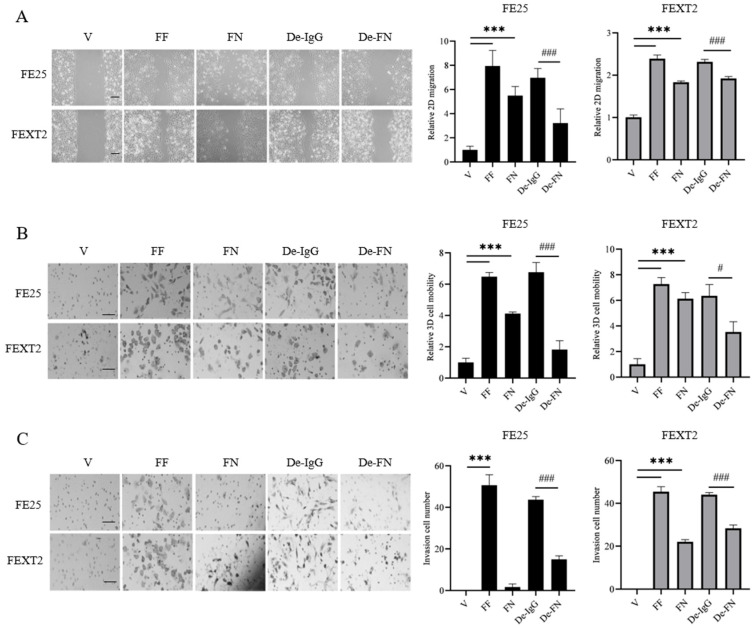
Depletion of FN partially compromised the migration and invasion promoted by FF. (**A**) In vitro scratch assay (2D migration) of FE25 cells and FEXT2 cells treated with vehicle, 10% FF, FN (21 μg/ml), 10% FF depleted with IgG (De-IgG) and 10% FF depleted with FN specific Ab (De-FN) for 24 h under serum-free conditions. (**B**) Transwell cell motility assay (3D migration) of 2 × 10^4^ cells loaded in the upper chamber under serum-free conditions. 24 h after the same treatment panel, the migrated cells was visualized by Giemsa staining. (**C**) Matrigel invasion assay using 1 × 10 ^4^ cells loaded in the upper chamber and cells invade through the Matrigel insert were counted in three random areas. Scale bar: 50 μm. Data were calculated with mean ± SD from triplicate experiments or more. # *p* < 0.05, ***/### *p* < 0.001, by two-tailed, unpaired Student’s *t*-test.

**Figure 3 cells-15-00080-f003:**
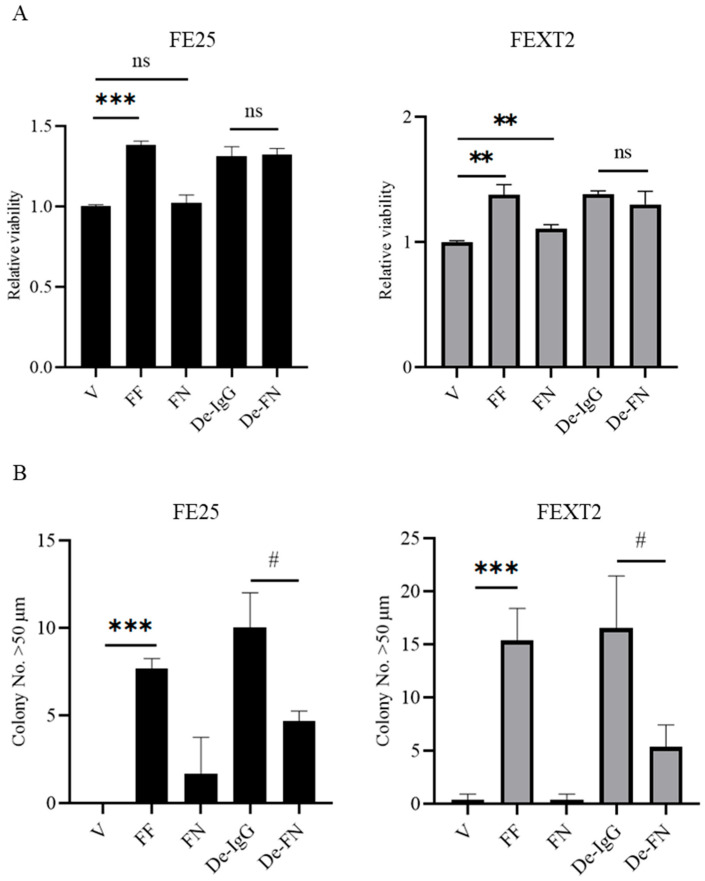
FN played little role in anoikis resistance and anchorage independent growth which were highly augmented by FF (**A**) In the modified anoikis assay, 4 × 10^3^ cells were incubated in a 3D suspension culture with 0.4% agarose on ultra-low attachment plates, and cell viability was assessed after 24 h using XTT colorimetric analysis. (**B**) Representative images of AIG colonies from both cell lines with identical pretreatments are shown. Colonies larger than 50 μm were counted after 14 days of culture. Data were calculated as mean ± SD from at least triplicate experiments. # *p* < 0.05, ** *p* < 0.01, *** *p* < 0.001, ns: not significant (*p* > 0.05), by two-sided unpaired Student’s *t*-test.

**Figure 4 cells-15-00080-f004:**
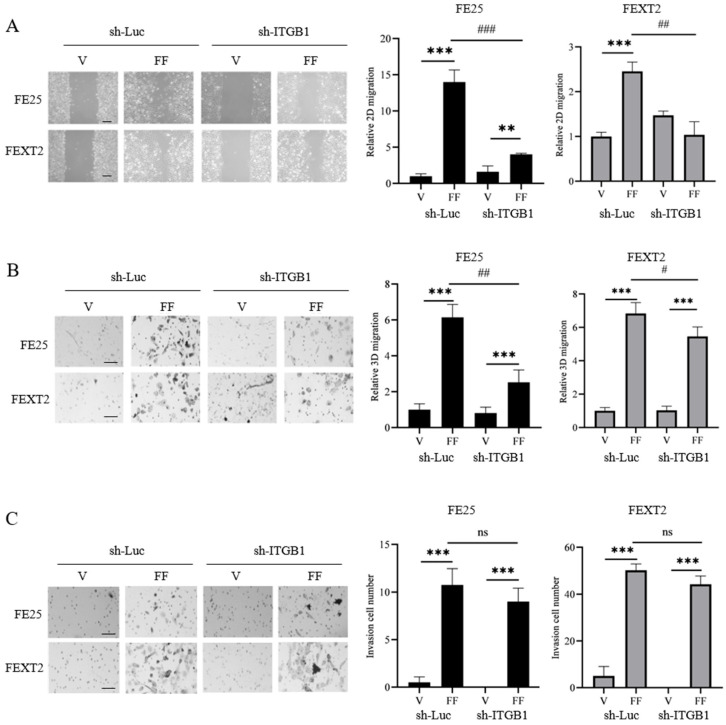
ITGB1 knockdown in FTE cells reduced 2D and 3D migration but not invasion of FTE cells in response to FF Under serum-free conditions, 2D (**A**) and 3D (**B**) cell migration assays and Matrigel invasion assay (**C**) were performed on vehicle- or 10% FF-treated FE25 and FEXT2 cells transduced with lentivector expressing short hairpin RNA against ITGB1 (sh-ITGB1) or control RNA (sh-Luc). Scale bar: 50 μm. Data were calculated as mean ± SD from at least triplicate experiments. # *p* < 0.05, **/## *p* < 0.01, ***/### *p* < 0.001, ns: not significant (*p* > 0.05), by two-tailed, unpaired Student’s *t*-test.

**Figure 5 cells-15-00080-f005:**
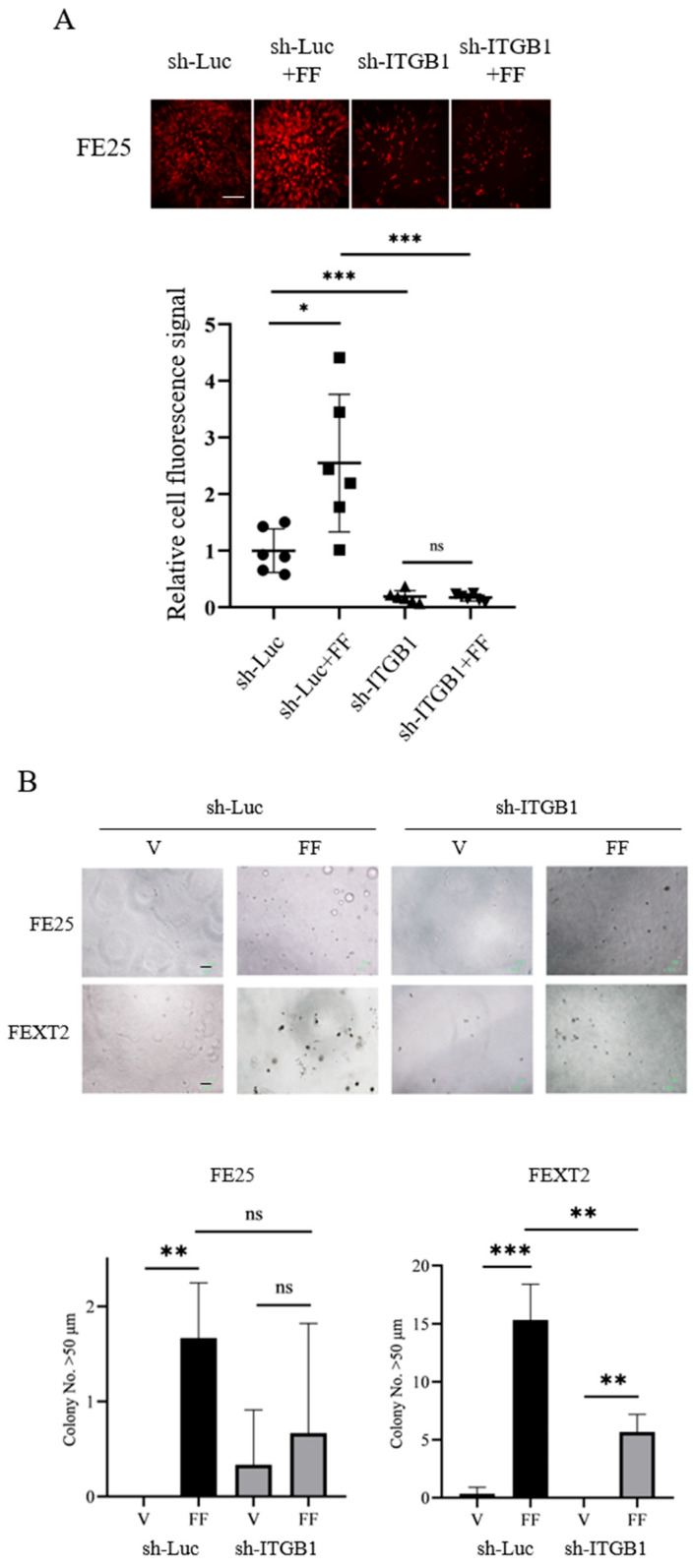
ITGB1 knock-down compromised peritoneal attachment growth and the AIG promoting effect of FF (**A**) The attachment of FE25-RFP cells (with or without ITGB1 knockdown) to ex vivo cultured-mouse peritoneal tissue was evaluated after treatment with either vehicle or 10% FF. Representative images display cell attachment after 24 h (scale bar: 200 μm). Quantification of adherent RFP cells was performed using fluorescence analysis in ImageJ (version 1.54p). Data from more than two independent experiments with triplicates were analyzed. Statistical significance: * *p* < 0.05, *** *p* < 0.001, determined by two-sided unpaired Student’s *t*-test. (**B**) AIG of FE25 and FEXT2 cells (with or without ITGB1 knockdown) was assessed in soft agar containing either vehicle or 10% FF, cultured for 14 days. Colonies larger than 50 μm were visualized and counted under a microscope. Data represent the mean ± SD from at least three independent experiments with triplicates. Scale bar: 100 μm. Statistical significance: ** *p* < 0.01, *** *p* < 0.001, ns: nonsignificant (*p* > 0.05), determined by two-tailed unpaired Student’s *t*-test.

**Figure 6 cells-15-00080-f006:**
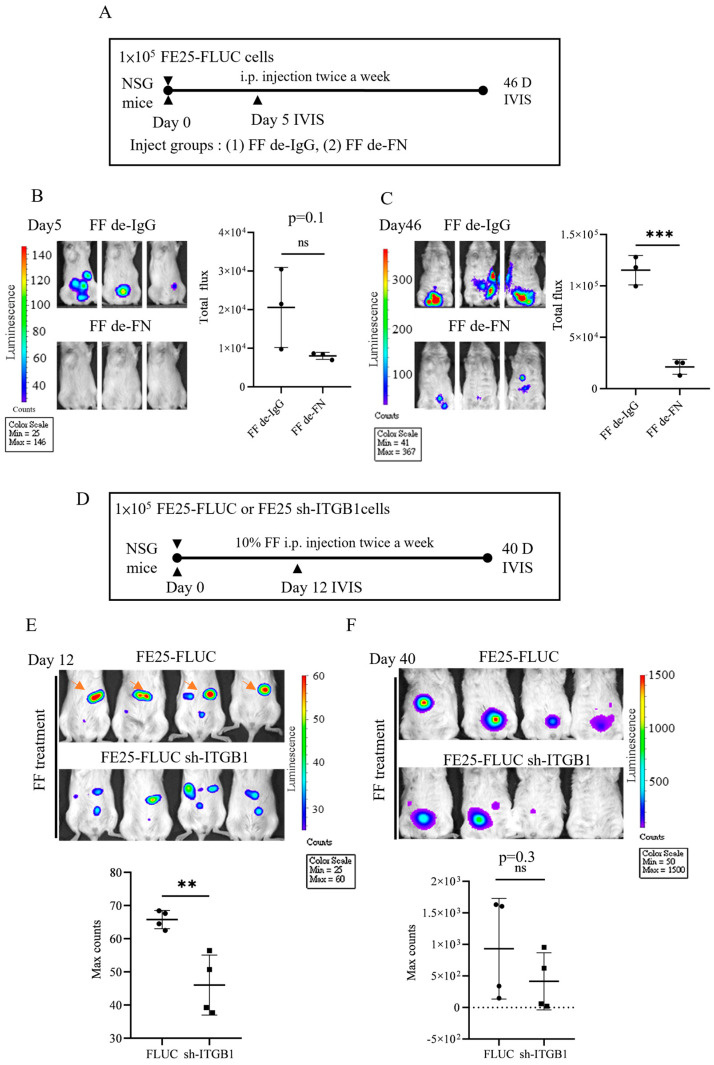
Depletion of FN in FF or knockdown of ITGB1 in FTE cells reduced the tumor-promoting activity of FF in intraperitoneal xenograft mouse model. (**A**) Luciferase expressing FE25 cells (1 × 10^5^) were injected intraperitoneally (i.p.) together with 10% FF depleted with IgG (De-IgG) or FN specific Ab (De-FN) into NSG mice. The treatments were repeated twice weekly until sacrifice on day 46. (**B**,**C**) On day 5 and day 46 mice, mice were subjected to IVIS to detect the viable cells. (**D**–**F**) The same FE25 cells (1 × 10^5^) with/without ITGB1 knockdown were i.p. injected together with 10% FF and same boosted in NSG mice. IVIS detection was performed on day 12 and day 40 used in both sets of experiments. Statistical results were analyzed using IVIS system software. ** *p* < 0.01, *** *p* < 0.001, ns: not significant (*p* > 0.05), by two-tailed unpaired Student’s *t*-test.

**Figure 7 cells-15-00080-f007:**
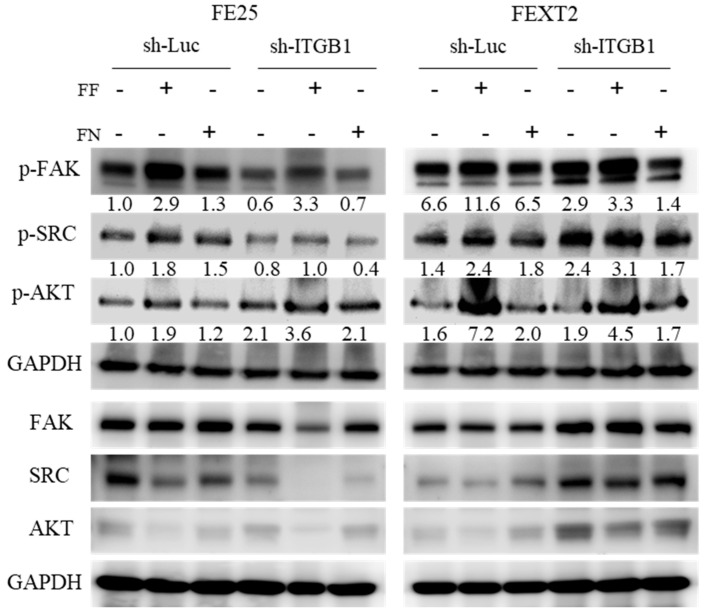
FF-FN activities were mediated by ITGB1-FAK/SRC/AKT pathways. After 30 min pretreatment with FF or FN, FE25 cells and FEXT2 cells with or without ITGB1 knock-down were subjected to Western blot analysis to detect phosphorylated FAK, SRC and AKT. Imaging was performed using a UVP multifunctional photographic film system, and band intensity was quantified by using Image J software.

**Figure 8 cells-15-00080-f008:**
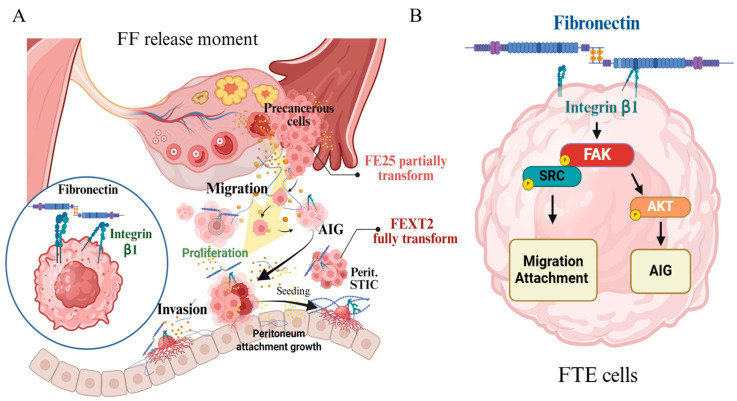
Summary of FF-FN mediated transformation activity and the signaling pathways in FTE cells (**A**) During ovulation, FF-FN is released into the peritoneal cavity, facilitating the migration of transforming FTE cells and enhancing their seeding and invasion within the peritoneal environment. Furthermore, the fully transformed FTE cells in the peritoneal cavity receive additional support from this same oncogenic source associated with ovulation. (**B**) Mechanistic studies indicate that in transforming FTE cells, FF-FN-ITGB1 signaling drives migration and attachment predominantly through the FN/ITGB1/FAK-SRC signaling. Additionally, ITGB1/FAK/AKT signaling promotes AIG. Collectively, these pathways are critical for FTE cell migration, attachment, and transformation, highlighting the central role of the FF-FN-ITGB1 axis in the early peritoneal dissemination of STIC and fallopian tube precancerous cells.

**Table 1 cells-15-00080-t001:** Role of FN in the transformed phenotype of FTE cells.

	Basal Level (Vehicle)	FF	FN	De-IgG	De-FN
**Proliferation**					
FE25	1.58	1.58×	1.08×	3.48	0.9×
FEXT2	1.95	1.43×	1.1×	2.37	0.9×
**Anchorage independent growth (>50 μm)**					
FE25	0	8 (No.)	1.2 (No.)	9.0	0.6
FEXT2	0.3	16.3 (No.)	0.3 (No.)	14.9	0.4
**Anoikis resistance (OD)**					
FE25	0.33	1.4×	1.0×	0.45	1.0×
FEXT2	0.39	1.4×	1.1×	0.53	0.9×
**Peritoneal attachment growth (ROI)**					
FE25	N/A	1.4×	N/A	1	0.8×
FEXT2	N/A	N/A	N/A	N/A	N/A
**2D Migration (Wound healing; µm^2^)**					
FE25	4.62 × 10^4^	8.8×	**5.5×**	3.36 × 10^5^	**0.5×**
FEXT2	2.3 × 10^5^	2.4×	**2.2×**	5.36 × 10^5^	**0.6×**
**3D Migration (Transwell assay; cell number)**					
FE25	5.75	6.1×	**4×**	38	0.3×
FEXT2	4.75	6.8×	**6×**	29.6	0.6×
**Matrigel invasion (Cell number)**					
FE25	0	54 (No.)	1.7 (No.)	44.3	0.3×
FEXT2	0	47 (No.)	**21** (No.)	45.3	**0.6**×

Scales of transformation phenotypes were: (1) Proliferation: Using XTT assay, (2) AIG: colony (>50 μm) number in soft agar, (3) Anoikis resistance: XTT assay in non- attached culture, (4) Attachment growth: fluorescence level of cells attached to mouse peritoneum, (5) 4-wells wound healing assay, (6) Cell mobility: number of cells migrated to lower part of transwell, (7) Matrigel invasion: number of cell in the lower part of matrigel transwell, (No.):expressed in quantity rather than in terms of ratio, N/A: no data is displayed.

**Table 2 cells-15-00080-t002:** Effect of ITGB1 knockdown on the transformation phenotypes of FTE cells.

Cell Line	FE25 sh-Luc		FE25 sh-ITGB1		FEXT2 sh-Luc		FEXT2 sh-ITGB1	
	Basal level (Vehicle)	FF	Basal level (Vehicle)	FF	Basal level (Vehicle)	FF	Basal level (Vehicle)	FF
**Proliferation**	0.27	**1.57×**	0.68	**1.18×**	0.86	1.91×	1.23	**0.99×**
**Anchorage independent growth (>50 μm, No.)**	0	2.0	0.3	0.7	0.3	15.3	0	**5.7**
**Anoikis resistance (OD)**	0.20	1.4×	0.20	1.4×	0.20	1.4×	0.25	2.0×
**Peritoneal attachment growth (ROI)**	1.16 × 10^5^	**2.5×**	2.2 × 10^4^	**0.17×**	N/A	N/A	N/A	N/A
**2D Migration (Wound healing, µm^2^)**	1.83 × 10^4^	**14.0×**	2.01 × 10^4^	**4×**	1.34 × 10^5^	2.4×	1.82 × 10^5^	**0.9×**
**3D Migration** **(Transwell assay, No.)**	5.25	**6.1**	0.8	**2.5**	8.75	6.8	1	**5.5**
**Matrigel invasion** **(No.)**	0.5	10	0	9	5	50	0	44.3

Scales of transformation phenotypes were: (1) Proliferation: Using XTT assay, (2) AIG: colony (>50 μm) number in soft agar, (3) Anoikis resistance: XTT assay in non- attached culture, (4) Attachment growth: fluorescence level of cells attached to mouse peritoneum (ROI), (5) 4-wells wound healing assay, wound healing level of the cell migration area, (6)Cell mobility: number of cells migrated to lower part of transwell, (7) Matrigel invasion: number of cell in the lower part of matrigel transwell, (No.):expressed in quantity rather than in terms of ratio, N/A: no data is displayed.

## Data Availability

All data generated or analyzed during this study are included in this published article [and its Appendix A].

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
