# Peer review of "Ovulation-Derived Fibronectin Promotes Peritoneal Seeding of High-Grade Serous Carcinoma Precursor Cells via Integrin β1 Signaling"

_cells, 2026, doi:10.3390/cells15010080_

Round 1
Reviewer 1 Report
Comments and Suggestions for Authors
The manuscript examines fibronectin effect on peritoneal seeding of high-grade serous carcinoma precursor cells by integrin signaling pathway.
Generally, the manuscript is well written.
Comments:
- Figure 1B: please add the vehicle data.
- Figure 1: compares FF with >100 KDa FF: please explain why >100 KDa FF lost significant effect compared to FF?
- Figure 1: please indicates p value is used to compare what?
- Figure 3A: on the right side, FEXT2, it seems the "**" labeled on the wrong area?
- Figure 6: The sample size is very small. Please explain.
- Figure 7: please provide the levels of total FAK, SRC, and AKT?
- Please list all abbreviations.
Reviewer 2 Report
Comments and Suggestions for Authors
This is a bit of a chaotic paper studying the role of fibronectin in ovarian cancer colonization of the peritoneum, that is, metastasis. The authors employ cell lines representing early- and late-stage ovarian cancer and observe that fibronectin, an abundant protein in follicular fluid, triggers cell migration, particularly of early precursors.
The paper starts by identifying fibronectin in FF (is it the fourth or the seventh most abundant protein in FF?). The authors demonstrate that depletion of fibronectin from FF reduces cell migration, again much more significantly in early precursors. Not surprisingly, integrin beta1 depletion reduced 2D and 3D migration in response to FF; but not Matrigel invasion. It also reduces peritoneal attachment in ex vivo models of peritoneal tissue and intraperitoneal xenograft growth. Unsurprisingly, FN attachment triggers adhesive signaling, including phosphorylation of FAK, SRC and AKT, which is abrogated in integrin beta1 knockdowns.
The paper is not bad but is poorly written and requires some controls.
Fibronectin has different functional domains, and it would be interesting to supplement depleted FF with fibronectin fragments to study integrin usage.
What are the functional beta1 integrins expressed in these cells? At least characterize the fibronectin receptors, alpha4beta1, alpha5beta1 and alphaVbeta3/5.
Along the same lines, can supplemented recombinant VCAM-1 compensate for the function of FN?
Please show controls of efficiency of beta1 depletion.
Reviewer 3 Report
Comments and Suggestions for Authors
So overall, this is quite an interesting and well-executed study. The authors tackle a biologically plausible and clinically relevant question — how ovulation-associated components of follicular fluid might contribute to early dissemination of HGSC precursors. The focus on fibronectin and the integrin β1 pathway makes sense, and the in vitro and in vivo experiments are reasonably consistent.
The introduction is clear but maybe a bit too long ,paragraphs 3 and 4 could be tightened; right now the narrative repeats the “precursor escape” idea several times. . Relating FN-integrin signaling to such intrinsic genomic drivers might make the story more integrated. adding a short contextual paragraph about the historical evolution of ovarian-cancer treatments , as discussed in “Cancer Treatments: Past, Present, and Future” , would help readers see where preventive targeting of FN-ITGB1 could fit clinically. It doesn’t need to be long, just a sentence or two about translational perspective.
the FN-depletion and ITGB1-knockdown data are convincing qualitatively, yet the variability between animals in the xenograft section is quite high. Maybe an independent repeat or pooling across cohorts could strengthen the statistics. Also, the FE25/FEXT2 model system is useful for distinguishing precursor versus transformed states, but I wonder whether including another independent FTE-derived line, or even primary cells, would confirm that this isn’t a cell-line-specific phenomenon. The authors might also consider a validation experiment using an ITGB1-neutralizing antibody instead of genetic knockdown, to confirm the phenotypic link more cleanly.
The mechanistic interpretation is logical , FN binding to ITGB1 activates FAK/SRC and AKT , but it might be worth tempering the claim that FN is the major factor. From the text, it’s clear other FF components like HGF and versican could act synergistically. Perhaps a multi-factor depletion experiment or some pathway-inhibition test (for example, FAK or SRC inhibitors) could further isolate the FN contribution. I also think the western blots in Figure 7 could use clearer normalization and molecular weight markers; as shown they look a bit cropped.
Conceptually, the “ovulation-FN-ITGB1 axis” is compelling, but the authors could expand the discussion to situate their findings in the broader landscape of ovarian cancer genomics. For instance, recent sequencing work such as “Identification of novel gene fusions in high-grade serous ovarian carcinoma: implications for tumorigenesis and targeted therapy” highlights that genomic rearrangements often cooperate with microenvironmental cues. Typos:
Figure 6 panels could be rearranged so that the in vitro and in vivo data appear sequentially; the current order feels slightly disjointed. In the methods, the concentrations and dilutions for the ELISA and migration assays are detailed (which is good), though in some places units flip between µg/mL and mg/mL (see Fig. 2 caption, worth checking. There are also minor English slips (“a achment” missing ‘t’ in several places, likely from PDF conversion) that should be fixed before final submission.
Round 2
Reviewer 1 Report
Comments and Suggestions for Authors
The authors answered my questions. No more comments.
Reviewer 2 Report
Comments and Suggestions for Authors
Thank you for addressing some of my comments. While some experiments I proposed were interesting, I understand the time and resource limitations, and the additions are the minimum I'd require. I have no additional questions.
Reviewer 3 Report
Comments and Suggestions for Authors
ok